# Development of a Mobile Open-Circuit Respiration Head Hood System for Measuring Gas Exchange in Camelids in the Andean Plateau

**DOI:** 10.3390/ani13061011

**Published:** 2023-03-10

**Authors:** Wilfredo Manuel Rios Rado, Paola Katherine Chipa Guillen, Dione Huamán Borda, Víctor Vélez Marroquín, José Ignacio Gere, Walter Orestes Antezana Julián, Carlos Fernández

**Affiliations:** 1Facultad de Ciencias Agrarias, Universidad Nacional de San Antonio Abad del Cusco, Av. de La Cultura 733, Cusco 921, Peru; 2Estación IVITA—Maranganí, Universidad Nacional Mayor de San Marcos (UNMSM), Cusco 08258, Peru; 3Institute for Animal Science and Technology, Universitat Politécnica de Valencia, 46022 Valencia, Spain; 4Unidad de Investigación y Desarrollo de las Ingenierías, Facultad Regional Buenos Aires, Universidad Tecnológica Nacional, Ciudad Autónoma Buenos Aires C1179AAQ, Argentina

**Keywords:** alpacas, emissions, head hood, maintenance requirement, methane

## Abstract

**Simple Summary:**

Alpacas are a species of great importance in the Andean Plateau, however, there are not enough studies evaluating their energy metabolism and greenhouse gas emissions. Considering this, an indirect calorimetry system (head hood) was developed to measure these components. Overall, the proposed respiration system performed well in terms of both accuracy and reliability of measurements. Results from the validation study confirmed that alpacas have a lower metabolic basal rate and therefore lower energy requirements for maintenance than other small ruminants.

**Abstract:**

Peru has the largest inventory of alpacas worldwide. Despite their importance as a source of net income for rural communities living at the Andean Plateau, data on energy requirements and methane (CH_4_) emissions for alpacas are particularly lacking. In 2019, the International Panel on Climate Change (IPCC; 2006, and Refinement 2019) outlined methods for estimating CH_4_ emissions from enteric fermentation and no methane (CH_4_) conversion factors were reported for camelids. IPCC has since updated its guidelines for estimating CH_4_ emissions from the enteric fermentation of livestock at a national scale. For greenhouse gas (GHG) inventory purposes, conversion factors were developed for ruminants but not for domestic South American camelids (SAC), with this category including alpacas. A mobile open-circuit respirometry system (head hood) for the rapid determination of CH_4_ and CO_2_ production, O_2_ consumption, and thereafter, heat production (HP) for camelids was built and validated. In addition, an experimental test with eight alpacas was conducted for validation purposes. The average HP measured by indirect calorimetry (respiratory quotient (RQ) method) was close to the average HP determined from the carbon–nitrogen balance (CN method); 402 kJ/kg BW^0.75^ and 398 kJ/kg BW^0.75^, respectively. Fasting HP was determined by the RQ method and 250 kJ/kg BW^0.75^ was obtained. The metabolizable energy requirement for maintenance (MEm) was calculated to be 323 kJ/kg BW^0.75^ with an efficiency of energy utilization of 77%. When intake was adjusted to zero energy retention by linear regression, the MEm requirement increased to 369 kJ/kg BW^0.75^ and the efficiency decreased up to 68%. The CH_4_ conversion factor (Y_m_) was 5.5% on average. Further research is required to gain a better understanding of the energy requirements and CH_4_ emissions of alpacas in conditions of the Andean Plateau and to quantify them with greater accuracy.

## 1. Introduction

Global warming is a great concern at present and greenhouse gas (GHG) emissions from agriculture contribute to climate change. Enteric methane (CH_4_) from ruminants contributes approximately 6% of the total global anthropogenic GHG emissions [1]. Energy metabolism trials have reported that CH_4_ losses accounted for 2% to 12% of the total gross energy (GE) consumed by ruminants [2]. However, data on CH_4_ emissions from camelids are scarce.

The International Panel on Climate Change [3] produced the *Guidelines for National Greenhouse Gas Inventories* outlining methods for estimating CH_4_ emissions from enteric fermentation. They estimated that camels with 570 kg of BW produced 46 kg/head per year [4] and alpacas with 65 kg of BW produced 8 kg/head per year [5]. A limited number of studies, only two, were considered in the *Guidelines for National Greenhouse Gas Inventories* and no CH_4_ conversion factor was calculated. The 2019 Refinement to the IPCC [3] maintained the same amount of enteric fermentation emission previously mentioned for camelids, with no new information added and no CH_4_ conversion factor presented.

Worldwide, the population of camelids is large and mainly focused in South American countries, with a total of 8,972,852 individuals [6]. Among the South American camelids, alpacas and llamas are domesticated species. The breeding of alpacas is the main source of income for communities living on the Andean Plateau, and information about energy requirements is almost non-existent. Camelid nutritional requirements generally have been based on information extrapolated from requirements for domesticated goats, sheep, and cattle [7]. Furthermore, there is a paucity of comparative camelid nutritional data, with nutrient requirements usually extrapolated from the llama to the alpaca.

From an animal performance perspective, energy is quantitatively the most important nutrient as it supports all physiological states and quantifies energy losses as CH_4_ is essential to improve performance and for emissions inventories. It is essential to acquire gas exchange equipment in South America in order to quantify the metabolic rate and enteric CH_4_ emissions of camelids. Alpacas are one of the camelid species whose largest population is found in Peru: 66% [8]. They are considered functional ruminants but taxonomically they belong to the Tylopoda suborder, unlike Ruminantia [9,10], with whom they present anatomical and functional differences. Ruminants have a stomach divided into four cavities and alpacas have a three-compartment stomach [11]. Currently, there are no facilities to measure the gas exchange of alpacas, including CH_4_, in Peru. For this reason, it is necessary to design, build, evaluate, and validate a precise gas exchange system for alpacas that allows us to evaluate continuous respiration measurements. Indirect calorimetry has been widely used for feed evaluation, for the determination of energy requirements of animals, and for CH_4_ emissions measurements [12].

Usually, indirect calorimeters are associated with high-cost facilities where respirometry chambers and equipment are allocated in a laboratory building. Due to the high cost, the objective of the present study was to build and validate a mobile open-circuit respirometry system (easy to transport and to be used in the field) with a respiration head-hood for camelids: alpacas, llamas, and vicuñas. During this first study, the respiration system will be used indoors. In the future, it is planned to put all the electronic equipment, the analyzer and the computer, in a cabinet protected from the external environment and use it in the field. Additionally, an experimental test was carried out with alpacas around maintenance, where heat production (HP) was determined by two methods: respiratory quotient (RQ method) and carbon-nitrogen balance (CN method).

## 2. Materials and Methods

### 2.1. Open-Circuit Respiration System

The gas exchange system presented in this study is based on the open-circuit respiration system developed by Fernández et al. [13]. This system was built and configurated for alpacas thus allowing continuous monitoring of O_2_, CO_2_, and CH_4_. The system consisted of a metabolic crate, a head hood, a computerized control, acquisition, and data recording system for gases and airflow and a gas cooler to remove moisture from the collected air sample. The instruments were installed on a mobile cart to make the system portable (Figure 1).

The stainless-steel pipes, the flowmeter (Thermal mass flowmeter, SensyMaster FMT450, ABB, Zurich, Switzerland), the gas cooler (Sample Gas Cooler, SCC-C, ABB, Zurich, Switzerland), and the fan (Helio Centrifugal Fans In-Line, TD-MIXVENT, Soler & Palau) were coupled at the bottom of the mobile system. Conversely, the rotameter (RATE-MASTER^®^ Flowmeter, RM SERIES, Dwyer Inc., Michigan City, IN, USA), the diaphragm pump (Diaphragm Pump 4N, ABB), the gas analyzer for the determination of methane (CH4), carbon dioxide (CO_2_), and oxygen (O_2_) (Continuous gas analyzers, EasyLine EL3020, ABB, Zürich, Switzerland), the data collection system (CR800 Series Dataloggers, Campbell Scientific Inc., Logan, UT, USA), and an integrated computer (PC Pos-D) for system control is located at the top. All analytical devices were purchased with digital output for connection to a computer. The entire system is connected by a high-pressure hose to the head hood where the animal is caged.

The head hood was made with a 1.5 mm galvanized plate (680 mm long × 1500 mm high × 500 mm wide; volume = 510 L). It was placed on the front structure of the metabolic cage which was designed to attach to the head hood. The hood has a transparent acrylic window (bolted and glued with silicone on the edge) at the front (500 mm long × 800 mm high) and a drawer (450 mm long × 650 mm high × 500 mm wide) with a handle to allocate food and water. The drawer was locked by four lateral locks situated on the front side and in the main body of the head hood. A line of rubber was placed on the edge of the drawer to obtain a better seal. The head hood has an opening (680 mm long and 1100 mm high) on the back with a funnel-shaped woven nylon curtain, with an adjustable hole for the animal’s neck, completely attached and glued to the head hood. It has a nylon cord through a fold around the edge, which allowed it to be adjusted and tied around the animal’s neck to create a vacuum environment and to prevent gas leakage. The chamber was attached to the front structure of the metabolic cage, thus maintaining the animal’s ability to perform its basic functions. Atmospheric air entered through a hole (internal diameter 25 mm) in the head hood, while the main suction line was at the top of the head hood. The analyzer performs 60 analyses per minute and the software records the average value over that period. We used a full day (24 h) of the empty chamber to estimate the baseline atmospheric values (background concentrations).

The operation of the respirometry system is further described. This system had two different but interconnected sampling lines. The main line drew air through a hood attached to a corrugated polyvinyl chloride tube (PVC; inner diameter 30 mm) equipped with an air filter to prevent contamination from atmospheric dust. Total air flow was measured by a mass flowmeter with a range from 0 to 10,000 L/h. A secondary line (inner diameter of 5 mm), which is located after the mass flowmeter, took gas samples from the main line (1 L/min) using a membrane pump attached to the rotameter and connected to the gas analysis unit.

The gas sample was filtered and then passed through the gas cooler to remove moisture before it enters into the gas analyzer. The gas analyzer measured three gases (CH_4_, CO_2_ and O_2_) in parallel and the CH_4_ and CO_2_ are measured using infrared principle with a range from 0–0.15 and 0–1.5%, respectively. The analysis of O_2_ works on the paramagnetic principle with a range from 18–21%. The paramagnetic O_2_ analyzer was equipped with an atmospheric compensation module to account for changes in atmospheric pressure. Regarding the system control, the software used to operate the system was developed by ABB.

The scheme of the system is represented in Figure 2.

### 2.2. Whole System Calibration

Zero and standard gas calibration were manually initiated on the gas analyzer and automatically conducted before each experimental run by the initial injection of pure N_2_ followed by a standard gas mixture (1% for CO_2_, 1500 ppm for CH_4_, and 10% for O_2_) until the gas levels were stabilized. During the measurements, temperature and humidity were monitored inside the chamber. System calibrations were performed by gas recovery tests. The first recovery test was carried out with the controlled release of CO_2_ inside the chamber. The total gas released was determined gravimetrically using a precision electronic balance (80 kg, 0.05 kg precision), according to Maclean and Tobin [14] recommendations. Gas recovery calculations were obtained from the relationship between the released gas and the amount detected by the measurement system (analyzer). The second recovery test was performed by burning ethanol for a certain time, as recommended for small chambers [15]; the weight loss of the ethanol container is associated with O_2_ consumption and CO_2_ emission for a combustion equation.

### 2.3. Calculations

Methane and CO_2_ production, and O_2_ consumption, were calculated by using the Haldane transformation as described by Aguilera and Prieto [16], except that no theoretical values for atmospheric CH_4_, CO_2_, and O_2_ concentrations were used. Before gas measurement, atmospheric air was sampled, and the gas concentration values (background) were used as a reference in calculations.

### 2.4. Experimental Test

#### 2.4.1. Animals and Feeding

A total of 8 Huacaya male alpacas of similar body weight (58.3 ± 3.30 kg BW) were used for measuring energy metabolism and CN balances. A diet based on alfalfa pellets and oat hay was used in this study in the proportion 30:70 (on a feed basis). The feeding level proposed in the present study (323 kJ/kg BW^0.75^) was close to the metabolizable energy for maintenance (MEm), reported by Roque et al. [17] (346 kJ/kg BW^0.75^). The daily ration was offered once at 08·00 h. The alpacas were housed in a proper room where they were isolated from environmental conditions outdoors. The area where the study was carried out is located at 4200 m above sea level, has an average temperature of 12.7 °C, and a relative humidity of 36% (dry Puna).

#### 2.4.2. Chemical Analysis

Representative samples of feeds, feces, and urine were collected and stored at −20 °C for later chemical analysis. Both feed and feces were dried in a convection oven at 60 °C for 48 h and then samples were ground to pass through a 1 mm sieve on a knife mill (KN 295 Knifetec™, Foss, Hillerød, Denmark). The urine collected underneath the metabolism crate was dried by lyophilization. Chemical analyses were conducted according to the methods of AOAC [18] for DM (no. 950.46), ash (no. 942.05), carbon C, nitrogen N, and crude protein (CP, no. 990.03). Gross energy (GE) content was determined using a calorimeter bomb (Parr 6400 Oxygen Bomb Calorimeter), following the procedure proposed by the manufacturer. Neutral detergent fiber analysis [19] was performed with a heat-stable α-amylase and sodium sulfite using the filter bag technique in an Ankom200 digestion unit (Ankom Technology Corp., Macedon, NY, USA) and expressed inclusive of residual ash (aNDF). The chemical composition of dietary ingredients used in this study is presented in Table 1.

#### 2.4.3. Energy and Carbon–Nitrogen Balances, and Heat Production Determination

Alpacas were allocated individually in metabolism crates. Upon completion of 21 days of adaptation period to experimental conditions, feed intake and total fecal and urine output were recorded daily for each alpaca over a 3-day period. Feces were collected in wire-screen baskets placed under the floor of the metabolism crates and urine was collected by gravity through a funnel into plastic buckets containing 100 mL 10% (*v*/*v*) of H_2_SO_4_. The acidification of urine was necessary to prevent microbial degradation and the loss of volatile ammonium. Before the gas exchange evaluations, the alpacas were placed in the head hood for short periods of time (four hours) for training them to get used to the enclosed space.

Body weight (PERUTRONIX SERIAL RS-232) was recorded at the beginning and the end of the experimental phase. Representative samples of food, feces, and urine were collected and stored at −20 °C for later chemical analysis. Metabolizable energy intake (MEI) was calculated as the difference between the GE intake and energy losses in feces, urine, and CH_4_ (with an energy equivalent value of 39.539 kJ/L CH_4_ [20]). After energy and C–N balances, gas exchange was measured sequentially for 24 h/alpaca.

Daily HP (kJ), determined by the RQ method, was calculated according to Brouwer [20] for O_2_ consumption, CO_2_ and CH_4_ productions (L), and urine-N (g) as:HP = 16.18 × O_2_ + 5.02 × CO_2_ − 2.17 × CH_4_ − 5.99 × urine-N

This method is termed the RQ method because it is based on the determination of the respiratory quotient (L CO_2_/LO_2_) and the retained energy (RE) was determined as the difference between the MEI and HP. Heat production was assessed by an independent study which was carried out with 5 alpacas subjected to fasting conditions for 3 days. A total of 20 gas measurements were performed during a period of 20 min for each alpaca.

Another indirect approach for measuring HP is by calculating the difference between ME and retained energy (RE). In this method, called the CN method, the C balance includes the measurement of carbon in the feed and that voided in feces, urine, CO_2_, and CH_4_, whereas the N balance is based on the measurements of nitrogen in feed, feces, and urine. The C balance provided the total amount of C retained in the body and the amount of C retained in fat was calculated by subtracting the amount of C retained in protein determined by the N balance. Retained energy was calculated according to Brouwer [20] from the C (g) and N (g) balance:RE = 51.8 × C − 19.4 × N

The MEm and their efficiency of use for maintenance (k_m_) were determined by two methods: 1. At zero RE and 2. Regression. According to AFRC [21], NRC [22], and Noziere et al. [23], the efficiency of ME for growth and fattening (k_f_ = 0.78 × q + 0.006; being q the diet metabolizability) was used to correct to zero energy retention. Therefore, the ME for fattening (MEf) was: MEf = RE/k_f_*,* MEm = MEI – MEf, and k_m_ = FHP/MEm. The regression method uses the following linear regression: RE = −a + b·MEI.

#### 2.4.4. Data Analysis

All statistical analyses were performed using R software (R Foundation for Statistical Computing, Vienna, Austria). In the experimental trial, the repeatability between repeated measurements of the same animal measured within the open-circuit respiratory system was estimated as a function of the variance components for animals and the residual variance by using the library SixSigma [24]. The discrepancy in measurements between the RQ and the CN method for assessing HP was calculated with the next index (NE) as:NE = [(HP_RQ_ − HP_CN_)/(MEI)] × 100

Analysis of variance (ANOVA) was conducted on the experimental data, with the alpaca as the experimental unit (using the stats package of R and the functions aov and lm, respectively). Mean values of HP obtained by both methods were compared by Fisher’s least significant difference test, and significance was set at *p* < 0.05. Finally, regression analyses were performed for establishing the relationship between HP obtained by RQ and CN methods and for predicting RE from MEI (function lm of R).

## 3. Results

Our results showed that during the recovery tests the gas concentrations reached an equilibrium concentration. For the first recovery test (controlled release of pure CO_2_) we obtain a progressive and linear accumulation of CO_2_ inside the chamber (R^2^ = 0.997). The measure value was 104 ± 2% (mean value ± SD). For the second test (burning ethanol), the measured values of recovery were 98.5 ± 4.7% for CO_2_ and 102.2 ± 4.7 for O_2_ (mean value ± SD) and are within the range that is considered acceptable for this methodology. To reach a standard criterion to correct the gas flow, we use the second recovery test. Although a recovery test with CH_4_ was not performed, the analytical method to detect this gas is similar to that of CO_2_ and, for this reason, the same recovery % value was used. The results from the repeatability analysis of CH_4_ and HP are shown in Table 2. The repeatability for CH_4_ and HP measurements was 70% and 60%, respectively.

The results from the repeatability analysis of CH_4_ and HP are shown in Table 2. The repeatability for CH_4_ and HP measurements was 70% and 60%, respectively.

In the RQ method, HP is calculated from measurements of O_2_ consumption, CO_2_ and CH_4_ production, and the amount of N excreted in the urine. Conversely, for the CN method, the HP is calculated as the difference between the ME and the RE as measured by the CN balances. The results for each individual alpaca from both HP methods used in the present study are presented in Table 3. No significant differences (*p* > 0.05) were obtained when HP was compared between methods. The HP from the RQ method was compared with the values obtained by the CN method and the regression analysis resulted in the following function:HP_RQ method_ = 1.009 ± 0.0016 (SE) × HP_CN method_
R^2^ = 0.998; RSE = 1.829; n = 8
where SE is the standard error, R^2^ is the coefficient of determination, and RSE is the residual standard error. The higher R^2^ and lower RSE indicated a good agreement between the values obtained by the RQ and CN methods. Figure 3 shows the closeness between the regression of the HP line (solid blue line) and Y = X line (dash line). Although the HP measurements obtained from both methods are proportional, the CN method tends to underestimate HP. These differences between methods are somehow expected due to variation in feed composition, feed level, and, consequently, deposition of energy in the body.

Evaluation of discrepancies in the estimation of HP determined by the RQ and the CN methods averaged 0.85%. Although both methods are not completely independent of each other, the close agreement found between them can be considered as an indicator of the absence of a systematic error. The FHP obtained during the experimental test was 250 kJ/kg BW^0.75^. The MEm was estimated by either zero energy retention or by regression. After correcting the energy balance to zero energy retention, the estimation of *k_m_* and MEm was 0.77 kJ/kg BW^0.75^ and 323 kJ/kg BW^0.75^, respectively. Using the data from the 8 experimental alpacas, the RE (kJ/kg BW^0.75^) was regressed against MEI (kJ/kg BW^0.75^), and the next simple linear regression was obtained:RE = −251.73 ± 21.038 + 0.682 ± 0.0444 × MEI

With RSE = 3.932 and R^2^ = 0.9752. Therefore, the FHP estimated by regression was the intercept and the value obtained was a rounded value of 252 kJ/kg BW^0.75^, and the slope of the equation represented a k_m_ of 0.68. Dividing FHP by k_m_ yielded a MEm of 369 kJ/kg BW^0.75^ (Figure 4).

## 4. Discussion

It is worth mentioning that this mobile respiration system is unique equipment in Peru and previously no gas exchange measurement studies have been carried out in SAC in their native land. The accuracy of the gas exchange determination is further dependent on the ability of the system to measure gas composition and, importantly, the total volume of the air moved through the respirometry system [13]. We observed that all of the calibration factors were very close to 1. This demonstrates the absence of leaks which was reflected in an accurate performance of the respirometry system with this being calibrated either gravimetrically or by ethanol burning.

Gage repeatability and reproducibility (Gage R & R) is a methodology used to define the amount of variation found in measurements obtained from an open-circuit calorimetry system. This compares the measurement variation with the total variability observed, consequently defining the capability of the measurement system. Measurement variation consists of two important factors: repeatability and reproducibility. Repeatability is related to equipment variation (indirect calorimeter device), whereas reproducibility is due to operator variation (the last does not apply in the present study). The repeatability for CH4 and HP measurements was 70% and 60%, respectively. The study by Robinson et al. [25] reported a repeatability of 79% and 81% for CH4 and HP, respectively. In another study by the same author [26] (both studies were conducted with sheep in portable accumulation chambers), the repeatability value agrees for CH4 (76%), but is lower for HP (60%). Studies using respiration chambers and sheep [27] found a repeatability of 89% for CH_4_. Oddy et al. [28], using the same method but with sheep in different physiological stages, found a repeatability value of 65% for CH_4_. Although the repeatability found in the present study is within the range reported in the literature, an improvement could be achieved by adjusting both CH_4_ and HP measurements by live weight and feed intake [25,26,27,28].

In this study, discrepancies in the estimation of HP determined by the RQ and the CN methods averaged 0.85%. This value is more than satisfactory when the considerable amount of both technical and analytical work is taking into account. Fernández et al. [13,29,30] obtained discrepancies of 0·005%, 0.055%, and 1.92% respectively, with goats and sheep using a face mask and a head hood. Aguilera and Prieto [16] reported a value of 0.018% in a study carried out with wethers caged in respirometry chambers.

Both RQ and CN methods are partially dependent on each other and the close agreement between them might be considered further evidence of the system’s reliability. The values obtained between methods are not identical [31,32], and our trial confirmed the RQ method systematically results in higher values of HP than the CN method (402 vs. 398 kJ/kg BW^0.75^ on average for RQ and CN methods, respectively). The CN method generally results in an overestimation of RE because the CN balance is usually overestimated due to evaporative and other losses in excreta [33]. The RE determined with the CN method was greater than that calculated with the RQ method (74 vs. 71 kJ/kg BW^0.75^, on average, respectively). Van Saun [34] reviewed nutrient requirements research on llamas and alpacas and that information was compiled at NRC [22]. He advised that in most cases, energy requirements for SAC were not available.

The FHP was measured in our study to compare the data obtained with the limited data available from the literature. Fasting reduces the possible effect of the diet on HP to a minimum. The good agreement with published data can be considered as further evidence of the system’s reliability. In our study, we found an FHP of 250 kJ/kg BW^0.75^ for alpacas, and no information was found previously in the literature. Camels have very low metabolic rates, particularly when dehydrated. The FHP was calculated as 215 kJ/kg BW^0.75^ and 166 kJ/kg BW^0.75^ for hydrated and dehydrated, respectively [35]. Environmental temperature also affects resting energy expenditure in llamas. The study by El-Nouty et al. [36] found an FHP of 467 kJ/kg BW^0.75^ in warm temperatures (27 °C) and 264 kJ/kg BW^0.75^ in hot temperatures (40 °C). The NRC [32] did not report values for FHP in camelids and, for sheep and goats, the committee provided a value of 296 and 332 kJ/kg BW^0.75^ for each species, respectively. Nielsen et al. [37] measured energy metabolism in llamas, sheep, and goats. Llamas in comparison with sheep and goats showed lower FHP (246 kJ/kg BW^0.75^ vs. 333 kJ/kg BW^0.75^ and 414 kJ/kg BW^0.75^, respectively). The FHP value found in our study with alpacas was pretty close to the value obtained with llamas by Nielsen et al. [37]. Large variation among species of camelids has been reported and it is well known that differences in FHP could be explained by the previous MEI of the animals. Fasting heat production is typically determined after a three-day fasting period and one might expect less variation in energy expenditure than in the fed state. Distribution of body components (fat and protein) is important in determining FHP, especially with *ad libitum* prior measurements of FHP. In general, it seems that camelids had a lower FHP than small ruminants, but this is largely conditioned by environmental conditions (as it affects the basal metabolic rate). In other studies, Vernet et al. [38] compared the energy utilization of three diets by llamas and sheep. The authors found that the energy expenditure was, on average, 22% lower in llamas than sheep fed straw. This finding is in agreement with Nielsen et al. [37] who found that llamas had 22% and 20% lower energy expenditure than sheep and goats fed straw, respectively. A direct energy metabolism comparison including alpacas was not found.

In this study, the values for FHP and MEm were estimated by linear regression. Obtained values were: 252 kJ/kg BW^0.75^ and 369 kJ/kg BW^0.75^, respectively. Guerouali et al. [39], in studies conducted in respiration chambers with dromedaries at rest, found that HP was 217 kJ of FHP/kg BW^0.75^ and the MEm was 305 kJ/kg BW^0.75^, respectively. In the same study, dromedaries utilized ME below maintenance with an efficiency of 73% and above MEm with an efficiency of 61%. Schmidt-Nielsen et al. [35] found a value of MEm of 217 kJ/kg BW^0.75^ with a mask when the animals were under heat stress conditions. Engelhardt and Schneider [40], using the CN balance technique, estimated that the MEm of the llama was 256 kJ/kg BW^0.75^, whereas Carmean et al. [41], using the respiration calorimetry technique, obtained a value of 353 kJ/kg BW^0.75^ as the MEm requirements for llamas. The NRC [22] suggested a MEm of 305 kJ/kg BW^0.75^. This value was derived from the mean of two estimates obtained from llamas. A slightly lower MEm value for alpacas (276 kJ/kg BW^0.75^) was reported by Newman and Paterson [42]. Literature showed variation in MEm due to different environmental temperatures (heat stress), gas exchange techniques or nutrition trials, diet, and plane of nutrition, which probably affected the estimation of MEm values. These ranged from 300 kJ/kg BW^0.75^ to 447 kJ/kg BW^0.75^ for sheep according to the level of activity and ME content of the diet and between 431 kJ/kg ^BW0.75^ and 456 kJ/kg ^BW0.75^ for goats [35]. Nielsen et al. [37] compared energy metabolism in llamas, sheep, and goats fed high-quality green grass hay and low-quality grass straw and found that llamas had lower daily MEm (328 kJ/kg BW^0.75^) than sheep (438 kJ/kg BW^0.75^) and goats (394 kJ/kg BW^0.75^). Greater variability in estimates of MEm vs. FHP is expected in camelids because of different levels of feeding and differences among diets and environmental conditions probably affect metabolic rate. The lower values of MEm and FHP found in alpacas when compared with other ruminants suggest that camelids have a lower basal metabolic rate than that reported for small ruminant species. This could be a consequence of adaptation to the harsh environment, where mechanisms of natural selection would have favored animals with superior ability to adapt to situations with a scarcity of feed [22,37].

Both IPCC [3] and FAO [43] guidelines outline methods for estimating CH_4_ production for inventory purposes from enteric fermentation and have developed some empirical equations for estimating total CH_4_ emissions for ruminants with total feed intake as the main driving factor (either on a DM or GE basis). Methane emissions have also been reported as a proportion of feed intake. For instance, the proportion of the GE intake which is lost as CH_4_ is known as the Y_m_ factor [3]. The IPCC Tier 2 methodology [3] suggests a Y_m_ value of 6.5% for large ruminants (without differentiating between species) and there are no specific recommendations for either small ruminants or camelids. The latest refinement of the IPCC guidelines displayed a variation of the Y_m_ values for large ruminants (ranging from 5.7 to 7.0%) and incorporated specific values for sheep (6.7%) and goats (5.5%). Nevertheless, emission values for camelids were not considered. The resulting value for Y_m_ found in the present study for alpacas fed a blend of alfalfa pellet and oat hay was 5.5% on average. Nielsen et al. [37] found that llamas in comparison with sheep and goats had lower CH_4_ emissions, when expressed as Y_m_: 4.0% vs. 5.3% and 6.9%, respectively. According to these authors, camelids had lower CH_4_ emission compared with true ruminants, probably due to the distinctive anatomical features allowing water-soluble nutrients to escape forestomach fermentation. Further research is required for a better understanding of the energy requirements of alpacas and to quantify their enteric CH_4_ emissions in the dry conditions of the Andean Plateau with greater reliability and accuracy.

## 5. Conclusions

The precision of the gas exchange was obtained by calibration factors demonstrating the absence of leaks and accurate performance of the respirometry system. The repeatability (amount of variation in measurement data with the respiration equipment) for CH_4_ and HP measurements was 65% on average, being in the range obtained in the literature. Discrepancies in the estimation of HP determined by the RQ and the CN methods averaged 0.85%, a rather satisfactory value that should be highlighted when accounting for the considerable amount of technical and analytical work involved. Therefore, the open-circuit indirect calorimetry system based on the head hood described in the present study is a useful tool for continuous monitoring of various components of energy metabolism and GHG emissions in alpacas. This system also allows for dynamic changes in these variables to be recorded over the course of a day or for longer periods. Due to the lack of bioenergetic information and CH_4_ emissions in camelids, the system described here is suitable and feasible for studies of gas exchange and energy metabolism under a wide range of physical, physiological, and nutritional situations.

## Figures and Tables

**Figure 1 animals-13-01011-f001:**
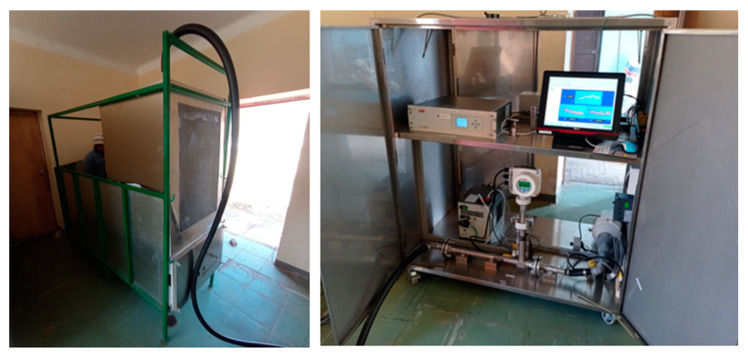
Mobile open-circuit respirometry system used for alpacas in the present study. The head hood is attached to the front structure of the metabolism crate.

**Figure 2 animals-13-01011-f002:**
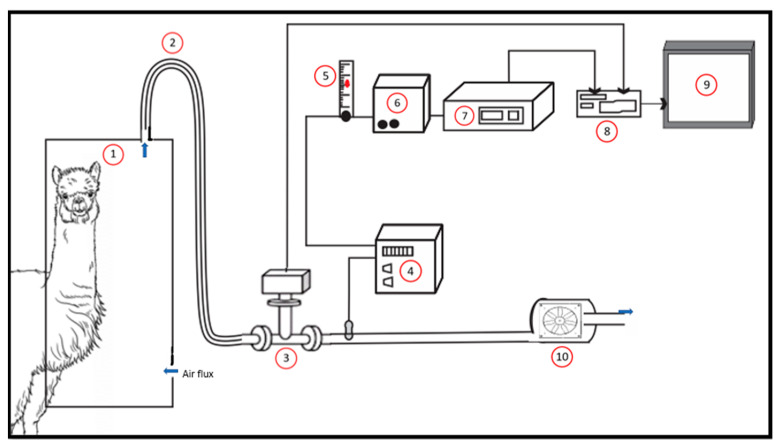
Schematic diagram of the mobile open-circuit respirometry system: (1) head hood, (2) high pressure hose, (3) flowmeter, (4) gas cooler, (5) rotameter, (6) diaphragm pump, (7) gas analyzer, (8) data logger, (9) integrated computer, and (10) fan.

**Figure 3 animals-13-01011-f003:**
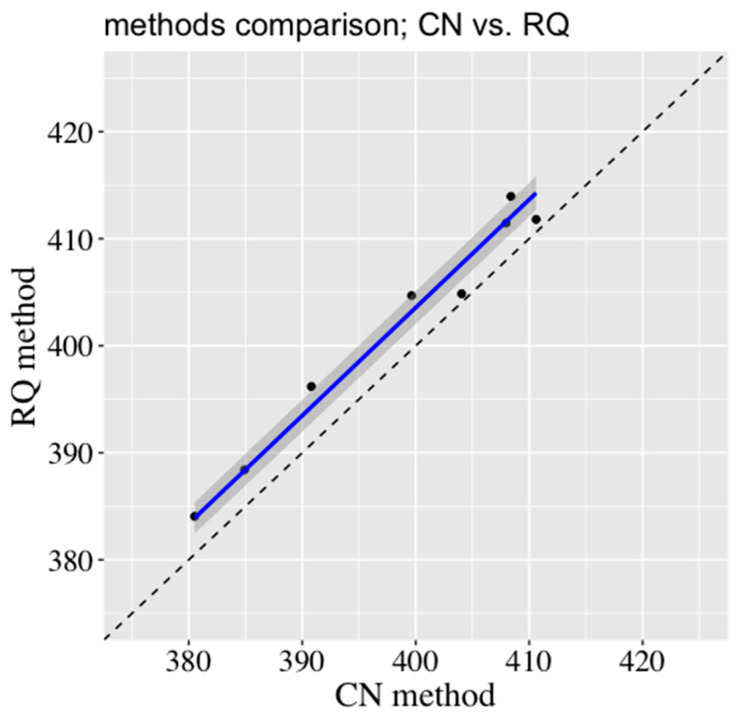
Regression analysis of HP determined by the RQ method in relation to HP measured from the CN method (solid line). Y = X is represented by a dashed line.

**Figure 4 animals-13-01011-f004:**
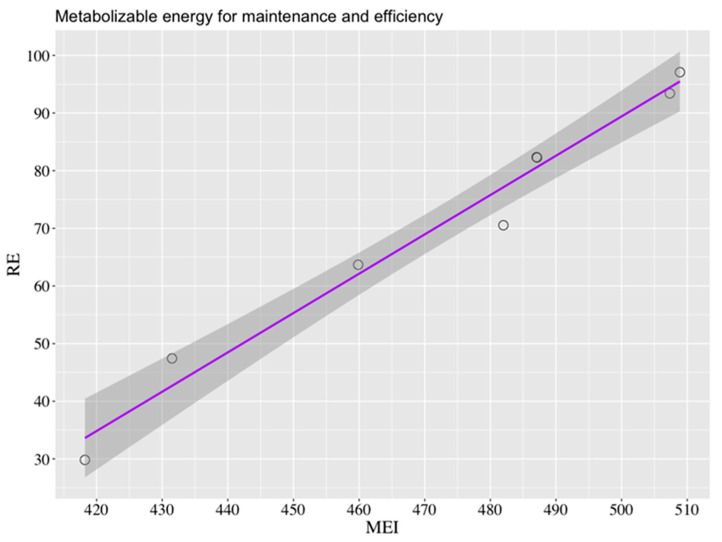
Regression analysis of RE over MEI expressed in kJ/kg BW^0.75^.

**Table 1 animals-13-01011-t001:** Ingredients and chemical composition (g/kg DM) of dietary ingredients.

Item	AP	OH
Ingredients, g/kg on a feed basis		
Alfalfa pellets	300	-
Oat hay	-	700
Chemical composition, g/kg		
DM	922	911
Ash	89.2	40.1
CP	170	52.8
GE ^1^	186	183
NDF ^2^	446	609
C	452	458
N	27.2	8.5

AP, alfalfa pellets; OH, oat hay; DM, dry matter; CP, crude protein; C, carbon; N, nitrogen. ^1^ GE: gross energy (kJ/kg DM). ^2^ NDF: assayed with a heat stable amylase and expressed inclusive of residual ash.

**Table 2 animals-13-01011-t002:** Repeatability study of the respiration system.

Item	CH_4_		HP ^1^	
Variation Measurement	VarComp ^2^	Contribution (%)	VarComp ^2^	Contribution (%)
Total Gage R&R	0.06	70	11,688	60
Repeatibility	0.06	70	11,688	60
Part to part	0.03	30	7792	40
Total variation	0.09	100	19,480	100

^1^ HP = heat production; ^2^ VarComp = variance component.

**Table 3 animals-13-01011-t003:** Energy partitioning (kJ/kg BW^0.75^) and carbon–nitrogen (g/kg BW^0.75^) balances of adult alpacas (n = 8) fed at a maintenance level of intake.

Item		Alpaca 1	Alpaca 2	Alpaca 3	Alpaca 4	Alpaca 5	Alpaca 6	Alpaca 7	Alpaca 8	Mean	S.E.M	CV%
	BW, kg	49	43	52	65	61	60	66	70	58.3	3.3	16
	DMI, kg/d	0.843	0.764	0.861	0.959	0.964	0.853	0.924	1.089	0.907	0.035	11
	CH_4,_ g/d	19	13	17	15	19	16	13	21	16.7	1.1	18
	CH_4_, g/kg DMI	23	18	19	15	20	18	14	20	18.4	1.0	15
	GE intake	834	835	811	771	815	728	731	822	793	15.7	6
	Energy in feces	297	274	269	236	268	261	256	254	264	6.3	7
	Energy in urine	20	9	13	13	11	8	12	13	12	1.3	29
	Energy in methane	57	44	47	35	49	40	31	48	44	2.9	19
	MEI	460	509	482	487	487	418	431	507	473	11.8	7
RQ method												
	O_2_ L/d	349	327	383	440	421	398	430	476	403	17.4	12
	CO_2_ L/d	352	330	371	435	407	390	399	484	396	17.1	12
	O_2_ L/d	27	19	23	20	27	22	18	30	23	1.5	19
	RQ	1.01	1.01	0.97	0.99	0.97	0.98	0.93	1.02	0.985	0.010	3
	HP	396	412	411	405	405	388	384	414	402	4.0	3
	RE ^1^	64	97	71	82	82	30	47	93	71	8.2	33
	Y_m_, %	6.8	5.3	5.8	4.6	6.0	5.6	4.3	5.9	5.5	0.290	15
CN method												
	C intake	20.70	20.74	20.14	19.15	20.25	18.08	18.17	20.43	19.71	0.387	6
	C in feces	7.53	6.97	6.85	5.68	6.53	6.42	6.25	6.36	6.57	0.195	8
	C in urine	0.517	0.221	0.326	0.331	0.302	0.208	0.320	0.354	0.322	0.033	29
	C in CO_2_	10.47	10.94	10.79	10.90	11.06	10.18	10.12	11.05	10.69	0.135	4
	C in CH_4_	0.775	0.599	0.642	0.480	0.664	0.549	0.423	0.657	0.599	0.040	19
	C retained	1.40	2.01	1.54	1.76	1.70	0.732	1.06	2.01	1.52	0.159	29
	N intake	0.644	0.644	0.630	0.584	0.611	0.537	0.534	0.625	0.601	0.016	7
	N feces	0.322	0.259	0.254	0.306	0.268	0.238	0.216	0.257	0.265	0.012	13
	N urine	0.150	0.081	0.094	0.095	0.093	0.062	0.118	0.112	0.101	0.009	26
	N retained	0.172	0.304	0.282	0.183	0.250	0.237	0.200	0.256	0.235	0.017	20
	RE ^2^	69	98	74	87	83	33	51	99	74	8.1	31
	HP ^3,4^	391	411	408	400	404	385	381	408	398	4.1	3

S.E.M, standard error of mean; CV, coefficient of variation; BW, body weight; DMI, dry matter intake; MEI, metabolizable energy intake; HP, heat production; RE, retention of energy; Ym, methane conversion factor; C, carbon; N, nitrogen. ^1^ Calculated as RE = MEI − HP; ^2^ Calculated as RE = 51.8 × C retained − 19.4 × N retained; ^3^ Calculated as HP = MEI − RE3; ^4^ HP comparison between methods (RQ vs. CN); non-significant at *p* < 0.05.

## Data Availability

Not applicable.

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
