# Peer review of "Development of a Mobile Open-Circuit Respiration Head Hood System for Measuring Gas Exchange in Camelids in the Andean Plateau"

_animals, 2023, doi:10.3390/ani13061011_

Round 1
Reviewer 1 Report
Introduction:
Introduction showed the need for information about entheric methane emission and energy partitionig of camelids.
Introduction also presented that “there are no specialized facilities for gas exchange measurement in alpacas”, but did not explain the reason for this need. Why camelids needs a specialized system?
The system developed is suitable for all kind of camelids or just for alpacas?
The study aim to “construct and validate a mobile (to use in the field) open circuit indirect calorimetry system”. I suggest change the term indirect calorimetry by respiration system, since the methane measurements are a concern too.
The system is considered mobile just because of the mobile cart? Aren’t necessary battery and a more compact and robust system for field measurements at the Andean? Should the external climatic conditions impact the respirantion exchanges measurements?
Materials and methods:
The system design is adequataly described. The figure and shematic draw is very helpful to understand the system’s operation.
The whole system calibration should be better described, with more details about the recovery tests performed.
The measurements of gases concentrations in the air from head hood and from ambient air (baseline) should be better described, for example, frequency and duration of each cycle.
It is necessary to describe how was performed the animal adaptation to the head hood system.
It is necessary to describe the method for fasting HP determination.
I suggest to presente the topic “Chemical analysis” after the description of C-N balances and heat production determinations.
A 24-h measurement time was enough for accuracy of respiration trial?
The CN method sould be better described.
Was the study submitted to animal care comittee?
I suggest the calculation of the coefficient of variation between alpacas and the repeatability of CH4 emission and HP (proportion of variance between animals with respect to the total variance).
I suggest the presentaion of the results of O2 consumption and CO2 production, as well the RQ.
Results
Lines 236-246 should not be located at the topic Results.
The results of the recovery test should be specific for each gas (O2, CO2 and CH4) analyzed, including mean±SD.
The results are not clearly presented. A major editing of style is required.
Dicussion
A major editing of style is required for the Discussion.
The conclusions are not supported by the results. The study aimed to develop and validate a portable respiration system for camelids. Conclusion based on the comparasion between metabolic rates of alpacas and small ruminants is not adequate.
Author Response
Response to Reviewer 1 Comments
Point 1: Introduction
Introduction showed the need for information about entheric methane emission and energy partitionig of camelids.
Introduction also presented that “there are no specialized facilities for gas exchange measurement in alpacas”, but did not explain the reason for this need. Why camelids needs a specialized system?
The system developed is suitable for all kind of camelids or just for alpacas?
The study aim to “construct and validate a mobile (to use in the field) open circuit indirect calorimetry system”.
Dear reviewer. Thank you very much for your comments. We have clarify. What we want to say is that there is no facilities to measure gas exchange in peru. Thank you
The equipment that measures gas is for all camelids, as indicated in line 86. In this study we have only worked with alpacas. To work with llamas, which are larger than alpacas, we only have to adjust the height of the drawer - cage. The rest is exactly the same
I suggest change the term indirect calorimetry by respiration system, since the methane measurements are a concern too.
Ok, we will change it throughout the text. We will keep it only in the introduction because the system we have created is based on indirect calorimetry. It is not a green feed type sensor system or sf6 type markers. We believe it is important to make clear the conceptual basis of the system at the beginning even though later, throughout the manuscript, we talk about the respiration system. Thank you
The system is considered mobile just because of the mobile cart? Aren’t necessary battery and a more compact and robust system for field measurements at the Andean? Should the external climatic conditions impact the respirantion exchanges measurements?
Dear reviewer, very good question. All equipment will allow movement. But currently he is working indoors. In the future it is expected to use it as a field tool. For this, all the electronic components, analyzer, computer, etc., will have to be placed in a cabinet with better protection from the outside. But that will be another phase. Thank you very much, good question. We will clarify it in the text.
Point 2: Materials and methods
The system design is adequataly described. The figure and shematic draw is very helpful to understand the system’s operation.
Thank you very much for your comments
The whole system calibration should be better described, with more details about the recovery tests performed.
We include more information about the whole system calibration. The recovery test performed are briefly described, and more details will be found in the bibliography cited
The measurements of gases concentrations in the air from head hood and from ambient air (baseline) should be better described, for example, frequency and duration of each cycle.
Done, thank you
It is necessary to describe how was performed the animal adaptation to the head hood system.
Done, thank you
It is necessary to describe the method for fasting HP determination.
Done, thank you
I suggest to presente the topic “Chemical analysis” after the description of C-N balances and heat production determinations.
Done, thank you
A 24-h measurement time was enough for accuracy of respiration trial?
Yes it is. But also depends on the objectives. There are works that only measure during a few hours. Other times the measurements are 2-3 hours after eating to see the evolution of methane. When a respiratory face-mask is used, animals cannot eat, then the measurement time is minutes. Since we are making daily determinations, we have made measurements during 24 hours. We have to find out the balance, associated with the objectives because the animal in a cage and in a head-hood for a long time, will surely reduce ingestion. This is without going into the problems of animal welfare and ethics committees in different countries
The CN method sould be better described.
Ok, yes, we will detail it more
Was the study submitted to animal care comittee?
Yes, it was, information about this is located at Institutional Review Board Statement segment on line 491
I suggest the calculation of the coefficient of variation between alpacas and the repeatability of CH4 emission and HP (proportion of variance between animals with respect to the total variance).
Thank you. We added already
I suggest the presentaion of the results of O2 consumption and CO2 production, as well the RQ.
We added already. Thanks
Point 3: Results
Lines 236-246 should not be located at the topic Results.
Done, thank you
The results of the recovery test should be specific for each gas (O2, CO2 and CH4) analyzed, including mean±SD.
Thank you. We added already
The results are not clearly presented. A major editing of style is required.
Done, thanks
Point 3: Discussions
A major editing of style is required for the Discussion.
The conclusions are not supported by the results. The study aimed to develop and validate a portable respiration system for camelids. Conclusion based on the comparasion between metabolic rates of alpacas and small ruminants is not adequate.
Ok you are right. We have re-written, thanks
Reviewer 2 Report
Comments to the Author
According to the manuscript, “Development of a mobile open-circuit respiration head hood system for measuring gas exchange in camelids at the Andean Plateau.” Even though the reviewer appreciates the authors' efforts in achieving the current experience, he nevertheless believes that the manuscript has several errors that prohibit it from being considered for publishing.
1. Introduction part, this research will contribute to the compilation of scientific information on the measuring of heat production using a head hood respiration chamber. The introduction section was almost thoroughly re-concisely detailed by focused on Alpaca energetic measurement.
2. Material and methods part, the experiment was conducted to evaluate the gas exchange and C-N balance. I am concerned about the lack of information on feed, fecal, and urine collection management; the animal was offered the same portion, and animal refinement might be included. Fat might be a component in animal feed composition. How can the C content be analyzed. Gas flow rate, sample duration, and calibration need to be explained for the system that is operating in open circuit. Prior to each test, the analyzers were calibrated using reference gases, which is a crucial step in the measurement process. The experimental methodology and hypothesis testing you decided to use to improve generalizability and response variability were not explained or justified by the author. Rearranging the table will be necessary.
3. Result and discussion section is required fundamental reconsideration. Even though these approaches are not entirely independent of each other, the statistical analysis may still be applicable. More information may be added to the discussion of energy partition. Sub-sections might be unnecessary.
4. The references section should be written in accordance with journal formatting instructions.
Author Response
Response to Reviewer 2 Comments
Point 1: Introduction
This research will contribute to the compilation of scientific information on the measuring of heat production using a head hood respiration chamber. The introduction section was almost thoroughly re-concisely detailed by focused on Alpaca energetic measurement.
Done, thank you
Point 2: Materials and methods
The experiment was conducted to evaluate the gas exchange and C-N balance. I am concerned about the lack of information on feed, fecal, and urine collection management; the animal was offered the same portion, and animal refinement might be included. Fat might be a component in animal feed composition. How can the C content be analyzed. Gas flow rate, sample duration, and calibration need to be explained for the system that is operating in open circuit. Prior to each test, the analyzers were calibrated using reference gases, which is a crucial step in the measurement process. The experimental methodology and hypothesis testing you decided to use to improve generalizability and response variability were not explained or justified by the author. Rearranging the table will be necessary.
We are completely agree. Ones we read again we found lack of information and meaning. So, we re-written all m&m. Thanks a lot
Point 3: Results and discussion
Result and discussion section is required fundamental reconsideration. Even though these approaches are not entirely independent of each other, the statistical analysis may still be applicable. More information may be added to the discussion of energy partition. Sub-sections might be unnecessary.
We have re-organized and re-written results and discussion. Even material and methods. It is true we found a difficult read. Thank you so much.
Point 4: References
The references section should be written in accordance with journal formatting instructions
Thanks for the observation, the references have already been modified according to the journal.

Reviewer 3 Report
This practical study is very complex and particularly interesting and useful, both for estimating the basal energy requirement of the alpacas species, and for evaluating the contribution of methane and carbon dioxide emissions to the global greenhouse effect. This research has a high level of originality, and the authors' approach to this research domain is just remarkable.
Overall, the manuscript is structured and edited skillfully. The introduction very clearly exposed the importance of this study, and emphasized the current gaps in the scientific documentation on this topic; the novelty of this research is properly pointed out.
Research methodology is accurately described. Interesting details are provided regarding the conceptualization, the key components of the experimental system and the investigation methods. The presentation of the research methodology is structured in well-defined sections and is accompanied by eloquent graphic images. The analysis/discussion of the results is professionally done by comparison with the results obtained by other researchers in gas exchange measurements, using various methods. It is worth noting that the authors took into account the behavior of the camelids likely affected by the investigation procedure; therefore, they considered it absolutely necessary to continue similar studies in natural living conditions in the Andean plateau. The theoretical calculation methods are clearly described.
The investigated bibliography is relevant but not very up-to-date, which I consider acceptable considering the originality of this study.
I found no any important issues to correct, except in the lines 332-333, where the verb should be added (or reformulate the phrase). Also, I recommend the authors to mention in the Conclusions a few words on the emissions of greenhouse gases from these species compared to other smaller ruminants. What about ammonia production in rumen fermentation? I think it should have been measured and taken into account when calculating N retained, for a better accuracy.
Author Response
Response to Reviewer 3 Comments
Point 1:
This practical study is very complex and particularly interesting and useful, both for estimating the basal energy requirement of the alpacas species, and for evaluating the contribution of methane and carbon dioxide emissions to the global greenhouse effect. This research has a high level of originality, and the authors' approach to this research domain is just remarkable.
Overall, the manuscript is structured and edited skillfully. The introduction very clearly exposed the importance of this study, and emphasized the current gaps in the scientific documentation on this topic; the novelty of this research is properly pointed out.
Research methodology is accurately described. Interesting details are provided regarding the conceptualization, the key components of the experimental system and the investigation methods. The presentation of the research methodology is structured in well-defined sections and is accompanied by eloquent graphic images. The analysis/discussion of the results is professionally done by comparison with the results obtained by other researchers in gas exchange measurements, using various methods. It is worth noting that the authors took into account the behavior of the camelids likely affected by the investigation procedure; therefore, they considered it absolutely necessary to continue similar studies in natural living conditions in the Andean plateau. The theoretical calculation methods are clearly described.
The investigated bibliography is relevant but not very up-to-date, which I consider acceptable considering the originality of this study.
I found no any important issues to correct, except in the lines 332-333, where the verb should be added (or reformulate the phrase). Also, I recommend the authors to mention in the Conclusions a few words on the emissions of greenhouse gases from these species compared to other smaller ruminants. What about ammonia production in rumen fermentation? I think it should have been measured and taken into account when calculating N retained, for a better accuracy.
Good point. Thanks a lot. But ammonia emission by rumen fermentation was not determined. Our analyzer only measure o2, co2 and ch4. But should be intereting in the future add an ammonia module to aour analyzer. Thanks again

Round 2
Reviewer 2 Report
The manuscript with the ID animals-2154603 needs to be revised with additional information.
- L423 The feeding level was close to the metabolizable energy for maintenance (MEm). It is necessary to cite the earlier study or citation source.
- Another consideration is that, depending on the methodologies used for data analysis, experimental design may provide more information about statistical models.
It appears that 40% of your manuscript incorporates plagiarism. Consider limiting it to less than 30%.
Author Response
Response to Reviewer 2 Comments
Point 1: L423 The feeding level was close to the metabolizable energy for maintenance (MEm). It is necessary to cite the earlier study or citation source.
Response 1: Done, thank you. Corrections were made on L184-L185
Point 2: Another consideration is that, depending on the methodologies used for data analysis, experimental design may provide more information about statistical models.
Response 2: The purpose of this study is to validate the equipement and the method for the determination of gas exchange in camelids, additional information is no relevant to the objetives of this study. Currently, the research group is developing studies about the energy metabolism in alpacas using the validated system.
Point 3: It appears that 40% of your manuscript incorporates plagiarism. Consider limiting it to less than 30%.
Response 3: We receive the plagiarism report of the software however we also note that most of the overlaps are common usage paraphrasing and should not be considered relevant to fix.
A more detailed revision in Ithenticate is necessary to correctly mark and reduce overlaps.
